# The Extraordinary Case of a Woman with a 30-Year-Long Diffuse Leishmaniasis Cured with One Single Ampoule of Intranasal Pentavalent Antimoniate

**DOI:** 10.3390/pathogens12070890

**Published:** 2023-06-29

**Authors:** Sheila V. C. B. Gonçalves, Dorcas L. Costa, João da J. Cantinho-Junior, José N. Vieira-Junior, Edna A. Y. Ishikawa, Rubens N. Costa, Antônio C. G. Costa-Filho, Ronald da C. Araújo, Silvia R. B. Uliana, Jenicer K. U. Y. Yasunaka, Adriano C. Coelho, Jackson M. L. Costa, Carlos H. N. Costa

**Affiliations:** 1Clínica Dermatológica, Hospital Getúlio Vargas, Teresina 64001-200, PI, Brazil; sheilacastelo@hotmail.com; 2Centro de Agravos Tropicais Emergentes e Negligenciados, Universidade Federal do Piauí, Teresina 64000-450, PI, Brazil; dorcas.lc@gmail.com; 3Hospital de Urgência de Teresina, Teresina 64017-775, PI, Brazil; cantinhojr@hotmail.com; 4Instituto de Doenças Tropicais Natan Portella, Teresina 64002-510, PI, Brazil; noronhajunior86@gmail.com; 5Núcleo de Medicina Tropical, Universidade Federal do Pará, Belém 66075-110, PA, Brazil; ishikawaufpa@gmail.com; 6Centro de Terapia Renal, Teresina 64003-075, PI, Brazil; rnerycosta@ig.com.br; 7Hospital Santa Maria, Teresina 64076-410, PI, Brazil; acgcf61@gmail.com; 8Departamento de Medicina Especializada, Universidade Federal do Piauí, Teresina 64049-550, PI, Brazil; ronaldcostaaraujo@uol.com.br; 9Departamento de Parasitologia, Instituto de Ciências Biomédicas, Universidade de São Paulo, São Paulo 05508-090, SP, Brazil; uliana.silvia@gmail.com (S.R.B.U.); jenicerk@usp.br (J.K.U.Y.Y.); 10Departamento de Biologia Animal, Instituto de Biologia, Universidade Estadual de Campinas, Campinas 13083-862, SP, Brazil; ccadriano@gmail.com; 11Centro de Pesquisas Gonçalo Moniz, Fundação Osvaldo Cruz, Salvador 40296-710, BA, Brazil; jackson.costa58@gmail.com

**Keywords:** diffuse cutaneous leishmaniasis, pentavalent antimonial, intranasal drug delivery, *Leishmania amazonensis*, CD8+ T-cells, anergy, immune cell exhaustion

## Abstract

Infection with *Leishmania amazonensis* and *L. mexicana* may lead to diffuse cutaneous leishmaniasis. The cure is exceptional, especially for the strange case of this lady. Case report: The patient acquired the disease in childhood and remained with lesions for over 30 years, albeit several treatments. She worsened after a pregnancy, developing disseminated lesions. Miltefosine with amphotericin B and pentamidine resulted in remission. Lesions reappeared after one year, accompanied by intra-nasal infiltration of the disease. The nasal spraying of a single ampoule of pentavalent antimoniate resulted in the sustained disappearance of the nasal symptoms and all the cutaneous lesions. After over eight years, she remains disease-free, albeit under renal replacement therapy. The high nasal mucosal antimonial concentration may explain the long-lasting cure via new MHC class I epitope-specific CD8+ cell clones against *L. amazonensis* present in the nasal mucosa.

## 1. Introduction

Leishmaniases are caused by protozoa belonging to the genus *Leishmania* and transmitted by the bite of sand flies. The infection leads to cutaneous, mucous, and visceral diseases. Cutaneous leishmaniasis, the most common presentation, is caused by several species of the subgenus *Viannia*, including *L. braziliensis* and *L. peruviana*, and by species of the *Leishmania* subgenus, such as *L. major*, *L. tropica*, and *L. aethiopica* in the Old World, and *L. mexicana* and *L. amazonensis* in the Americas. Most infections by *L. amazonensis* and *L. mexicana* in the New World and *L. aethiopica* in Africa [1] cause localized cutaneous leishmaniasis (LCL), usually responsive to treatment or even spontaneous cure. However, some patients infected by *L. amazonensis, L. mexicana*, or *L. aethiopica* develop a disseminated, anergic form called diffuse cutaneous leishmaniasis (DCL) that is difficult to cure, leading to extensive skin deformities that span decades [2]. Here, we present a very severe case of a woman who carried DCL for 30 years, since childhood, and had a striking disappearance of the illness after unprecedented treatment. The patient agreed to the publication of her clinical report and photographs and signed the informed consent form.

## 2. Case Report

This article addresses the medical evolution of a woman from the city of Urbano Santos, Maranhão State, Brazil. At five years of age, in 1985, she presented a small nodular infiltrative lesion in the left lower limb. The LCL diagnosis was confirmed by skin biopsy and successful treatment with meglumine antimoniate (C7H18NO8Sb) (Glucantime™). About four months later, similar lesions appeared on the lower limbs. The patient was again treated with pentavalent antimoniate but without improvement. Lesions spread to the upper limbs, trunk, nostrils, and ears. Delayed type hypersensitivity reaction (DTH), measured by the leishmanin skin test, was non-reactive. Histopathological examination revealed a diffuse intense histiocytic chronic inflammatory process with many amastigotes. The species were identified through a monoclonal antibody panel, revealing *L. amazonensis*. The patient was thus diagnosed with DCL (Figure 1).

As the disease evolved, all body segments were affected: upper limbs, lower limbs, trunk, face, oral mucosa, and, at the end of the disease, just before possible cure, the nasal mucosa. The entire skin was affected except for the scalp, palm plantar, and inguinocrural regions. From 1990 to 2002, the patient spent most of her life in hospitals receiving pentavalent antimoniate (Sb^V^), pentamidine isethionate, aminosidine sulfate, a combination of Sb^V^ with gamma-interferon, Sb^V^ associated with paromomycin sulfate and amphotericin B, and antibiotics. Initial improvement was obtained, but the lesions recurred after a few months. She received saquinavir and Imunoglucan™ once. She evolved with episodes of worsening, having sporadic treatments with liposomal amphotericin B when lesions were exacerbated. There was a significant disease worsening after her last pregnancy in 2009. At this time, she was treated with liposomal amphotericin B for 14 days and continued secondary prophylaxis with weekly doses.

In October 2010, soon after the delivery, a severe aggravation of the lesions was noted. Nodules and ulcers were widespread, and lesions appeared in the palate and nasal mucosa (Figure 2). Again, she received several courses of liposomal amphotericin B cycles. The renal function worsened with creatinine rising to ~2.5 mg/dL. A kidney biopsy showed glomerular deposits of IgM and C3, suggesting a progressive renal disease caused by drug therapy. HIV serology was negative, as were also an anti-nuclear factor, p-ANCA, and c-ANCA. Two isolates obtained in June and December 2008 were submitted to drug resistance screening, and no resistance was detected (Appendix A) [3].

In July 2012, she started combination therapy with miltefosine and liposomal amphotericin B (three months, with intervals), and two months later, with pentamidine (nine doses). She experienced a dramatic clinical improvement so that no active lesions and no parasites were seen in biopsies two months after the end of this therapy. However, one year later, the skin lesions returned and were full of parasites, although with a denser lymphocyte infiltrate (Figure 3).

The numerous therapeutic regimens were followed by progressive impairment of renal function, evidenced in 2013 by creatinine levels of 3.7 mg/dL, BUN of 161 mg/dL, and anemia leading to erythropoietin, limiting treatment alternatives.

The latest manifestation was the reappearance of lesions in the nasal mucosa, papules on the face, and nodules on her knees at the end of 2014. At that time, she complained of nasal obstruction, and the rhinoscopy showed bilateral thickening of the nasal turbinate with abundant purulent discharge (Figure 4). Due to renal disease, she was not getting systemic or local treatment then. However, as salvage therapy to alleviate the nasal obstruction, she started to be treated in January 2015 with intra-nasal pentavalent antimonial (Glucantime™—5 mL with 405 mg SV^5^, 81 mg/mL, made by Sanofi-Aventis Farmacêutica Ltda., Sao Paulo, Brazil), a single ampoule transferred to an appropriate flask for nasal use, sprayed in both nostrils twice a day for three weeks. The patient denied any discomfort with the treatment. Very surprisingly, after using only one vial, the nasal lesions were soon healed, and the cutaneous lesions on the face and knees, with total regression of all skin lesions (Figure 5). All images were published with the patient’s signed informed consent.

Since then, aside from scars, no new lesions have shown up. However, renal failure progressed, and the patient started renal replacement therapy in early 2017. At the end of 2018, a polymerase chain reaction from a skin snip revealed no parasite DNA, but DTH remained negative. She got an extensive skin allergic reaction at the beginning of 2022. A skin biopsy was undertaken, no parasites were found, and she recovered entirely. She regularly goes to outpatient dermatology consultations, and, at this moment (May 2023), more than eight years after the treatment with intranasal Glucantime™ spray, she does not have any new lesions. Meanwhile, she waits for her opportunity of having a kidney transplant.

## 3. Discussion

The case presented here is typical of DCL, which usually begins in childhood and is clinically characterized by skin-colored or erythematous papules, plaques, or nodules. Unlike LCL, ulceration usually does not occur. The disease evolves with the appearance of other similar lesions in the surrounding skin. Over months to years, hematogenous dissemination of the parasite occurs with other lesions’ appearance in various body parts. As with the patient, the most affected areas are the face and limbs, and the disease spares the scalp, armpits, and inguinocrural and palmoplantar regions. Infiltration may involve large areas of the body and, when present on the face, give the patient the leonine appearance, confounding with Virchow’s leprosy, like this lady [4,5,6].

In DCL, vacuolated macrophages are full of parasites, but no lymphoplasmacytic infiltrate, as happened with this patient. In addition, in CL caused by *L. amazonensis,* fewer patients have a reactive skin reaction to *Leishmania* antigens and immunohistochemistry, and ex vivo assays indicate the predominance of regulatory T-cell response leading to anergy. In DCL, the cellular immune response with lymphoplasmacytic infiltrate is absent. These findings of infections caused by *L. amazonensis* in LCL and DCL oppose the exuberant Th1-like cellular response in LCL caused by *L. braziliensis*. These aspects indicate that the primary pathogenic element of the patient’s DCL is immunosuppression [6,7].

Thus, the origin of DCL seems to be the disability of patients infected with *L. amazonensis, L. mexicana,* and *L. aethiopica* to mount a protective cell-mediated type immune response or the parasite’s ability to induce a regulatory response. The role of the host response in the development of DCL, in this case, is highlighted by the devastating worsening of the patient during and after pregnancy due to pregnancy immunotolerance [8]. However, analysis of the cellular response to isolates of *L. aethiopica* originating from patients with and without DCL suggested that the parasites have mechanisms for suppressing the cellular immune response [9]. Nevertheless, it is unknown which virulence factors are these. The vacuolated macrophages seen on CL and DCL by *L. amazonensis* resemble Virchow cells, seen in lepromatous leprosy [10], and may have similar immunosuppressive mechanisms through interference with lipid metabolism. Additionally, a pathway implicated in arginase I expression and enzymes involved in prostaglandin and polyamine synthesis, and where TGF-β has been demonstrated, results in a permissive environment for *L. amazonensis* progression to chronic disease [11].

The lack of response to leishmaniasis treatment may be due to either treatment failure or drug resistance itself [12]. However, the extensive response failure of *L. amazonensis* in this woman’s case to the various unrelated drugs suggests a lack of response to treatment rather than drug resistance. This hypothesis was confirmed by susceptibility testing against different drugs [3] and this work, which did not reveal significant differences between the isolates obtained from this patient and *L. amazonensis-type* strains. This fact suggests that manipulation of the host response by *L. amazonensis* has been the predominant factor in this patient’s failed attempts to treat DCL. Markers of CD4+ and CD8+ T-cell apoptosis from patients with DCL caused by *L. mexicana* and *L. amazonensis* have been described [13], suggesting that T-cell exhaustion is related to the maintenance of the disease [14,15].

However, the most enigmatic fact, in this case, was the unexpected sudden clinical and parasitological response to the last rescue treatment. Although DCL caused by *L. amazonensis*, *L. mexicana*, and *L. aethiopica* is curable, albeit challenging [16,17,18], this patient was an extreme case: numerous and prolonged treatments had failed, and she was cured after a single ampoule of Sb^V^ applied topically on the nasal mucosa. If resistance mechanisms to antimoniates existed, like lower drug accumulation, lower drug internalization, lower trypanothione concentration, and a lower rate of reduction of Sb^V^ to the trivalent form (Sb^III^), they were overcome [16]. Although its precise mechanism of action is unknown, a possible explanation for the treatment success was linked to the direct action of Sb^V^ on parasites present in the nostrils where mucosa drug absorption is fast and reaches high local concentrations [19]. However, the patient had already used dozens of ampoules of parenteral Sb^V^. These facts indicate that the distant action of Sb^V^ on the far skin site of one single ampoule was unlikely.

Therefore, questions remain about the reason for healing all skin lesions after the drug application of one ampoule in the nostrils. As the modulation of the immune response has been reported in the cure of two cases with immunotherapy, it may also have occurred in this case [7]. To this end, it is plausible that hidden antigen-specific epitopes would have been revealed to T-lymphocyte clones [9] after massive parasite death by local application of the Sb^V^, reversing anergy and enabling the patient to mount an effective response [15,20]. As DTH remained absent, CD4+ exhaustion reversal can be ruled out, and the recovery of immune response for parasite clearance must have been towards peptide epitopes presented by MHC I molecules, revealed through the action of a high concentration of Sb^V^, and recognized by CD8+ cells [21], finally overturning the tolerance modulated by exhaustion [22].

Additionally, as skin immunity is extended to the mucosa, it is plausible that the opposite may have happened, e.g., mucosa immunity had been extended to the skin [23], hence the clearance and cure of skin lesions. If this interpretation holds, T-cell CD8+ priming by previously hidden antigens signalizes that mucosal CD8+ immune response may be the cornerstone for developing *Leishmania* vaccines. From this perspective, we may anticipate that the use of Sb^V^ in the nostrils of patients without local *Leishmania* infiltration may not be effective. Local therapy in parasitized skin lesions had been tried in this patient, except it was not with Sb^V^ but with heat, and in the same way the parasites were killed, it also indeed killed the immune cells. However, if intralesional Sb^V^ in skin lesions of DCL caused by *L. amazonensis* would have a similar effect as in the mucosa, it would be a matter of wonder.

Since the patient is a candidate for kidney transplantation and immunosuppressive therapy, the future will tell if the apparent cure has sterilized *L. amazonensis* or if immunosuppression will reactivate the disease and rekindle her suffering. Finally, since medicine and scientific logic cannot dispute the metaphysical interpretation of phenomena, the patient’s last encounter revealed the different dimensions of incantation between doctors and patients. She interrupted unequivocally with the difficulty of finding and expressing a scientifically coherent explanation for her healing and her future: “But it is so simple Doctor, don’t you understand? It was and will be a miracle”.

## Figures and Tables

**Figure 1 pathogens-12-00890-f001:**
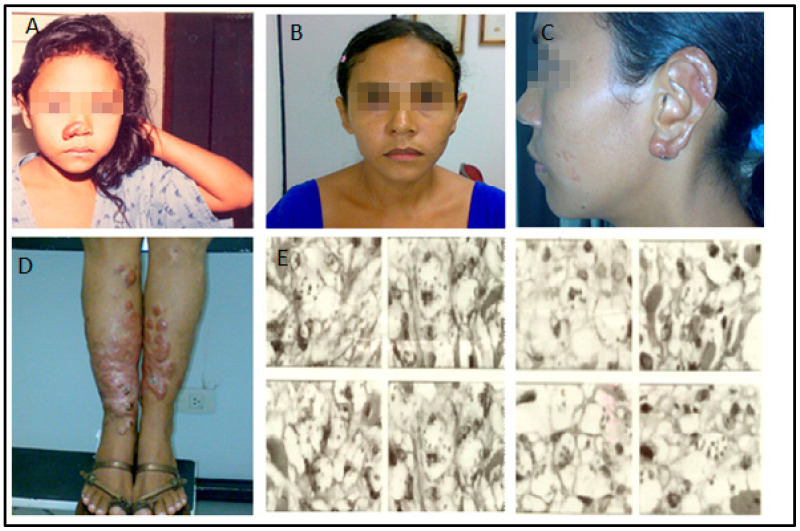
(**A**): The patient, in 1991, five years after the onset of the disease, recurrence of the nodule-infiltrative lesion in the right wing of the nose. (**B**–**D**): Disease progression (January 2008). Observe the infiltration in the ears and legs, but the face was still spared by this time. (**E**): Anergic leishmaniasis demonstrated by skin biopsy (December 2003). Note the presence of amastigotes, ballooning of infected histocytes, and absence of inflammatory infiltrate.

**Figure 2 pathogens-12-00890-f002:**
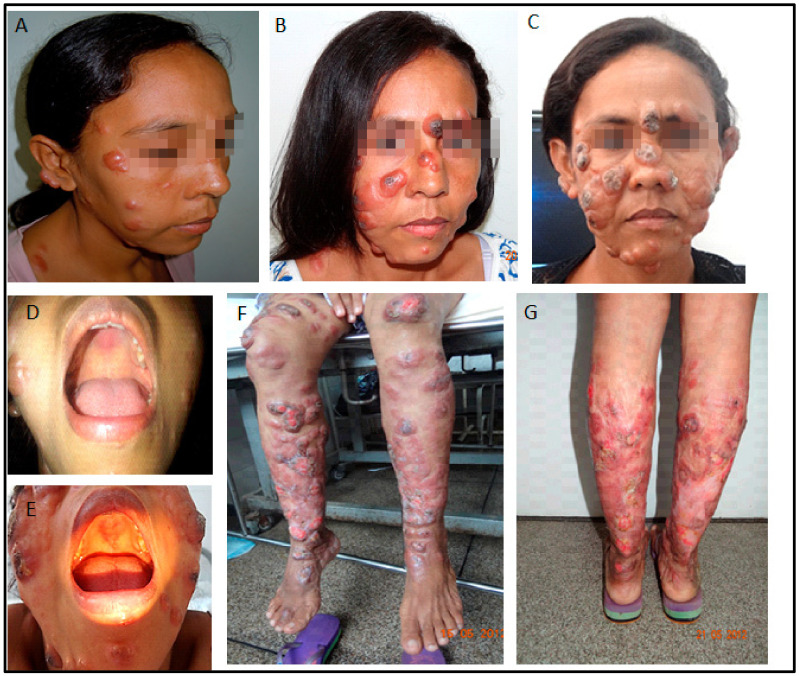
(**A**–**C**): Disfiguring ulcerative facial nodules (January 2011, October 2011, May 2012), 3 months, 12 months, and 1 month after delivery. (**D**,**E**): Mucosal palate involvement (2011, 2012). (**F**,**G**): Extensive lesions on the extremities in 2012, shortly before treatment with miltefosine and liposomal amphotericin B. Note the impressive worsening of lesions involving the face, extremities, and soft palate.

**Figure 3 pathogens-12-00890-f003:**
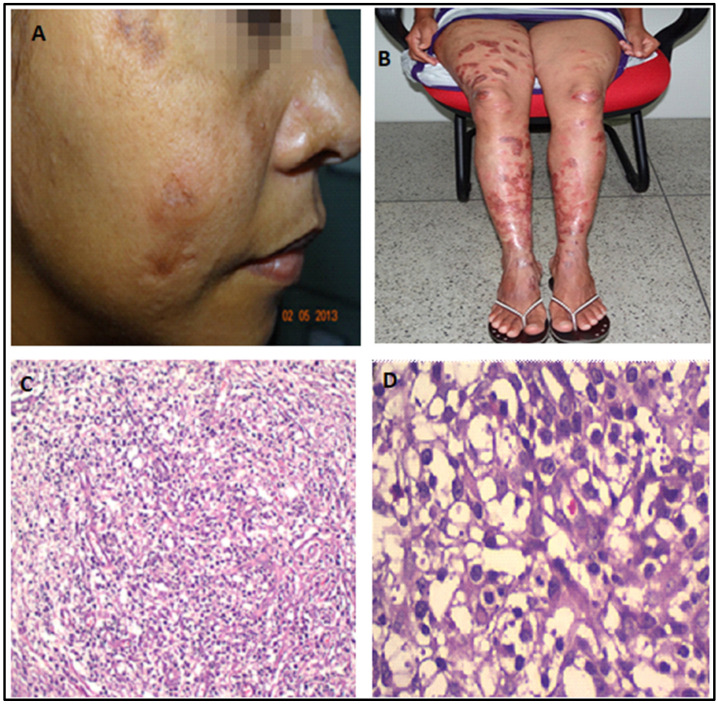
(**A**,**B**): Dramatic healing of lesions one month after initiating miltefosine and liposomal amphotericin B (May 2013). (**C**,**D**): Skin biopsy (2013). Interestingly, despite the clinical improvement, the skin biopsy still showed amastigotes, histiocytes, and a denser infiltrate of lymphocytes, similar to the biopsy performed in 2003.

**Figure 4 pathogens-12-00890-f004:**
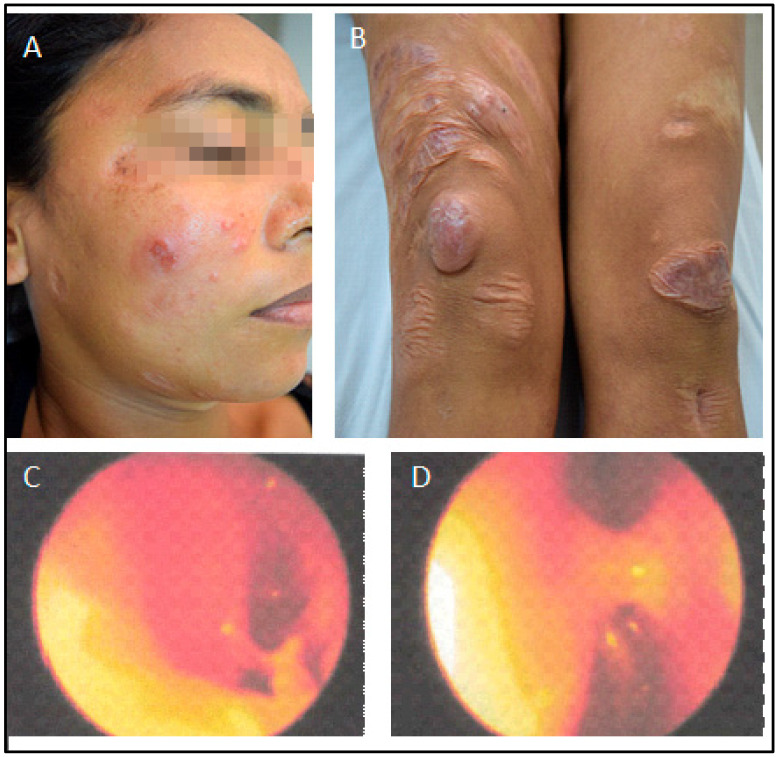
Skin and nasal situation before treatment with intranasal pentavalent antimonial (January 2015). (**A**,**B**). Papules in the face and nodules in the knees. Rhinoscopy. (**C**). Left nasal cavity. (**D**). Right nasal cavity showing nasal septum hyperemia with abundant purulent discharge.

**Figure 5 pathogens-12-00890-f005:**
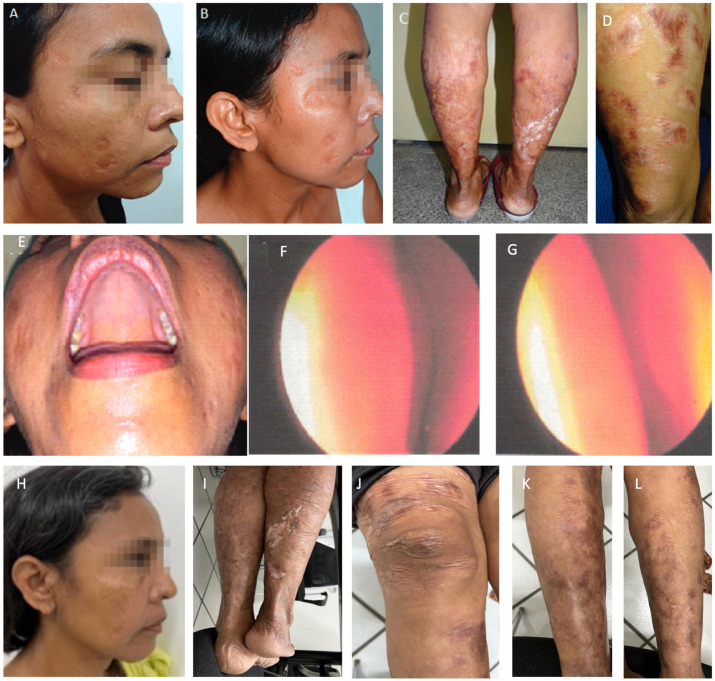
Patient after intranasal antimonial treatment. The first photo (**A**) is from July 2018, three years after the treatment, and photos (**B**–**G**) are from September 2019, four years after. (**E**): Full resolution of palate lesion following intranasal antimonial. (**F**): Rhinoscopy after intranasal antimonial treatment. Left nasal cavity; (**G**): Right nasal cavity: note minor edema, hyperemia, and absence of pathological secretion. Photos (**H**–**L**) were taken in May 2023, eight years after intranasal spraying of antimonial. Note sustained recovery and spread scars, but no inflammatory signs.

## Data Availability

The data is available will be provided by the corresponding author.

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
