# Peer review of "The Extraordinary Case of a Woman with a 30-Year-Long Diffuse Leishmaniasis Cured with One Single Ampoule of Intranasal Pentavalent Antimoniate"

_pathogens, 2023, doi:10.3390/pathogens12070890_

Round 1

Reviewer 1 Report

 It is better to use meglumine antimoniate (C7H18NO8Sb) for Glucantime, Glucantime is meglumine antimoniate as it is mentioned in the company’s brochure, although N-methylglucamine antimoniate is a Synonyms of meglumine antimoniate but usually MA is used.

Treatment of the patient needs more detail if available eg. The dose and duration of pentavalent antimoniate.

 References need to be improved as an example ref no 1 is about old world leishmaniasis caused by L. aethiopica and is not related to new world leishmaniasis.

I am wondering if the authors can include a more recent photos and more up to date information about the patient would be excellent.  

Author Response

Dear, thank you very much for your appropriate comments. Please my answers point-by-point.

Comment 1:  It is better to use meglumine antimoniate (C7H18NO8Sb) for Glucantime, Glucantime is meglumine antimoniate as it is mentioned in the company’s brochure, although N-methylglucamine antimoniate is a Synonyms of meglumine antimoniate but usually MA is used.

Answer 1: Thanks for the opportune suggestion. We made the change as recommended.

Comment 2: Treatment of the patient needs more detail if available eg. The dose and duration of pentavalent antimoniate.

Answer 2: Since the patient lived in other cities and was treated in different hospitals, we do not have a precise record of the duration and dosing of each treatment. However. We one of us (JMLC), who treated her earlier had the general registers described in the article.

Comment 3:  References need to be improved as example ref no 1 is about old world leishmaniasis caused by L. aethiopica and is not related to new world leishmaniasis.

Answer 3: Although L. aethiopica also causes diffuse leishmaniasis, we exchanged the cited reference, for a Technical Report by WHO as suggested by reviewer 2. (Now REF. 1).

Comment 4: I am wondering if the authors can include more recent photos and more up-to-date information about the patient would be excellent.  

Answer 4: Excellent remark, thanks. We provided photos taken in May, 2023.

Reviewer 2 Report

This is a thoroughly fascinating and excellent case presentation with fantastic image material collected over the years and a very well researched discussion. We must be grateful to the authors for this write-up. The introduction is useful but it is advisable to use an authoritative reference such as the WHO Technical Report Series 949 (2010) on leishmaniasis. It will be good to mention that DCL can also be caused by L. mexicana. It would be helpful to know when and whether the patient received conventional or liposomal amphotericin B, this was not always specified. It also was not specified in the description of treatment whether the patient received topical therapy at any point apart from the nasal spray, for example, was the first lesion she experienced treated with systemic or locally applied antimonials? From the conclusion we can derive that at no point the lesions were treated locally but it would be helpful to describe this earlier.   

Although the English language is exemplary there are some small edits that could be made, for example the first sentence of the abstract could be changed into: INFECTION WITH with Leishmania amazonensis may LEAD TO diffuse cutaneous leishmaniasis. Under figure 1: sentence should be changed to: Initial improvement was obtained but THE LESIONS recurred after a few months. And also: Again, she received several COURSES OF amphotericin B. 

Author Response

Dear, thank you very much for your appropriate comments. Please my answers point-by-point.

Comment 1: The introduction is useful but it is advisable to use an authoritative reference such as the WHO Technical Report Series 949 (2010) on leishmaniasis.

Answer 1: Thanks for this comment. We added this really authoritative reference.

Comment 2: It will be good to mention that DCL can also be caused by L. mexicana.

Answer 2: Now we made it clearer in the text. Please check it.

Comment 3: It would be helpful to know when and whether the patient received conventional or liposomal amphotericin B, this was not always specified.

Answer 3: Yes, you are right. However, we do not have the details of treatments previous to 2002, when she started to be treated in our hospital, as commented in response to reviewer 1.

Comment 4: It also was not specified in the description of treatment whether the patient received topical therapy at any point apart from the nasal spray, for example, was the first lesion she experienced treated with systemic or locally applied antimonials? From the conclusion we can derive that at no point the lesions were treated locally but it would be helpful to describe this earlier.   

Answer 4: The patient never took local drug therapy before the intranasal treatment. However, we used thermotherapy in some lesions during her worst phase, without response, in 2010.

Comments on the Quality of English Language

Comment 5: Although the English language is exemplary there are some small edits that could be made, for example, the first sentence of the abstract could be changed into: INFECTION WITH with Leishmania amazonensis may LEAD TO diffuse cutaneous leishmaniasis.

Answer 5: Correction performed.

Comment 6: Under Figure 1: the sentence should be changed to: Initial improvement was obtained but THE LESIONS recurred after a few months. And also: Again, she received several COURSES OF amphotericin B. 

Answer 6: Correction performed.

We made the indicated mistakes, used an automated English revision, and asked an English translator friend to review the manuscript.